# Minor Glomerular Abnormalities are Associated with Deterioration of Long-Term Kidney Function and Mitochondrial Injury

**DOI:** 10.3390/jcm9010033

**Published:** 2019-12-22

**Authors:** Byung Chul Yu, Nam-Jun Cho, Samel Park, Hyoungnae Kim, Hyo-Wook Gil, Eun Young Lee, Soon Hyo Kwon, Jin Seok Jeon, Hyunjin Noh, Dong Cheol Han, Ahrim Moon, Su Jung Park, Jin Kuk Kim, Seung Duk Hwang, Soo Jeong Choi, Moo Yong Park

**Affiliations:** 1Division of Nephrology, Department of Internal Medicine, Soonchunhyang University Bucheon Hospital, 170 Jomaru-ro, Bucheon 14584, Korea; nephroybc@schmc.ac.kr (B.C.Y.); ripley227@schmc.ac.kr (S.J.P.); medkjk@schmc.ac.kr (J.K.K.); sd7hwang@schmc.ac.kr (S.D.H.); 2Division of Nephrology, Department of Internal Medicine, Soonchunhyang University Cheonan Hospital, 31, Suncheonhyang 6-gil, Dongnam-gu, Cheonan 31151, Korea; c100086@schmc.ac.kr (N.-J.C.); samelpark17@schmc.ac.kr (S.P.); hwgil@schmc.ac.kr (H.-W.G.); eylee@schmc.ac.kr (E.Y.L.); 3Division of Nephrology, Department of Internal Medicine, Soonchunhyang University Seoul Hospital, 59, Daesagwan-ro, Yongsan-gu, Seoul 04401, Korea; hkim@schmc.ac.kr (H.K.); ksoonhyo@schmc.ac.kr (S.H.K.); jeonjs@schmc.ac.kr (J.S.J.); nohneph@schmc.ac.kr (H.N.); handc@schmc.ac.kr (D.C.H.); 4Department of Pathology, Soonchunhyang University Bucheon Hospital, 170 Jomaru-ro, Bucheon 14584, Korea; armoon@schmc.ac.kr

**Keywords:** glomerular filtration rate, glomerulonephritis, minor glomerular abnormalities, mitochondrial injury, urinary mitochondrial DNA

## Abstract

Minor glomerular abnormalities (MGAs) are unclassified glomerular lesions indicated by the presence of minor structural abnormalities that are insufficient for a specific pathological diagnosis. The long-term clinical outcomes and pathogenesis have not been examined. We hypothesized that MGAs would be associated with the deterioration of long-term kidney function and increased urinary mitochondrial DNA (mtDNA) copy numbers. We retrospectively enrolled patients with MGAs, age-/sex-/estimated glomerular filtration rate (eGFR)-matched patients with immunoglobulin A nephropathy (IgAN), and similarly matched healthy controls (MHCs; *n* = 49 each). We analyzed the time × group interaction effects of the eGFR and compared mean annual eGFR decline rates between the groups. We prospectively enrolled patients with MGAs, age- and sex-matched patients with IgAN, and MHCs (*n* = 15 each) and compared their urinary mtDNA copy numbers. Compared to the MHC group, the MGA and IgAN groups displayed differences in the time × group effects of eGFR, higher mean annual rates of eGFR decline, and higher urinary mtDNA copy numbers; however, these groups did not significantly differ from each other. The results indicate that MGAs are associated with deteriorating long-term kidney function, and mitochondrial injury, despite few additional pathological changes. We suggest that clinicians conduct close long-term follow-up of patients with MGAs.

## 1. Introduction

Minor glomerular abnormalities (MGAs) are unclassified glomerular lesions indicated by the presence of minor structural abnormalities by light microscopy, immunofluorescence, or electron microscopy that are insufficient for specific pathological diagnosis. They are frequently detected in patients with persistent, asymptomatic, and isolated proteinuria or microhematuria. MGAs are common and are found in 5.5% [1] of adults undergoing kidney biopsy to identify a specific pathologic diagnosis for constant isolated proteinuria, and 15.4% to 25.0% of patients with isolated microscopic hematuria [1,2]. 

Although many patients are diagnosed with MGAs after kidney biopsy, little is known about their impact on human health, and there are no clear definitions or guidelines regarding their management. To date, there are two case reports [3,4] and one retrospective study describing the pathology of MGAs [5]. Patients with MGAs generally follow a benign clinical course; however, the pathogenesis and long-term clinical outcomes of MGAs have not been examined. 

The kidney is an organ with high energy demands and a consequent abundance of mitochondria. Mitochondrial injury plays an important role in the pathogenesis of various kidney diseases, including acute kidney injury [6], chronic kidney disease [7,8,9,10], obesity-related hyperfiltration [11], and glomerulonephritis (GN) [12,13,14,15,16]. Recently, we demonstrated that mitochondrial injury is involved in early-stage glomerular inflammation prior to pathological changes in immunoglobulin A nephropathy (IgAN) by measuring urinary mitochondrial DNA (mtDNA) [17], which is used as a surrogate marker of mitochondrial injury in various kidney diseases [18,19,20]. Based on this finding, we hypothesized that mitochondrial injury may exist prior to noticeable pathological changes in MGAs, and that MGAs are associated with the deterioration of kidney function. 

To investigate this hypothesis, we retrospectively analyzed changes in kidney function and prospectively examined urinary mtDNA copy numbers in biopsy-proven MGA patients compared to matched healthy controls (MHCs) and patients with IgAN, which is the most common cause of primary GN worldwide. Our results indicated that MGAs are associated with deterioration of kidney function and elevated urinary mtDNA copy numbers.

## 2. Materials and Methods

### 2.1. Study Populations

This study was conducted according to the principles expressed in the Declaration of Helsinki. Clinical patient data were obtained from electronic medical records with the approval of the Institutional Review Board our hospital’s ethics committee (IRB no. 2016-01-002-007). 

To evaluate long-term kidney function, we retrospectively screened patients who had undergone kidney biopsy at Soonchunhyang University Bucheon Hospital between February 2001 and January 2017. Of the 1,097 patients who underwent renal biopsy, 72 (6.6%) were diagnosed with MGAs. An additional pathologist who had not performed the initial diagnoses reviewed kidney tissues from these patients and confirmed the presence of MGAs. We included MGAs patients whose kidney tissue contained more than 10 glomeruli according to light microscopy and at least 1 well-preserved glomerulus according to immunofluorescence and electron microscopy. In these pathological exams, we reviewed the entire kidney structure, including the glomerulus, tubulointerstitium, and microvasculature. When any borderline lesions were discovered that were suspected to be specific pathological diagnoses, the patient was excluded. We specifically excluded patients with > 5% interstitial fibrosis, tubular atrophy, and interstitial inflammation. In terms of vascular damage, we ruled out moderate to severe atherosclerosis and arteriolar hyalinization. We then carefully reviewed disease possibilities with minimal change disease, IgAN, and thin basement membrane disease, which may show minimal pathological changes. After exclusion of 23 patients who were followed for <1 year, 49 patients with MGAs were enrolled (Figure 1). We also enrolled age-/sex-/estimated glomerular filtration rate (eGFR)-matched patients with IgAN and MHCs (*n* = 49 each). MHCs were included if they had no history of diabetes mellitus (DM), hypertension (HTN), congestive heart failure, coronary artery disease, liver cirrhosis, or stroke, and were excluded if they had hematuria or proteinuria as diagnosed by urine dipstick testing and microscopy.

To evaluate mitochondrial injury, we prospectively enrolled patients with biopsy-proven MGAs (*n* = 15) from 279 biopsy-proven GN patients enrolled in the Cohort for Biomarker Inquiry of Renal Aggravation (COBRA) at Soonchunhyang University Seoul, Bucheon, and Cheonan Hospitals from May 2016 to January 2018. Age- and sex-matched patients with IgAN and MHC were also included (*n* = 15 each). All prospectively enrolled participants provided written informed consent.

### 2.2. Data Collection

We obtained demographic and comorbidity data, including a history of diabetes mellitus (DM) or hypertension (HTN). We collected body weight, height, body mass index, and systolic, diastolic, and mean arterial blood pressure (SBP, DBP, and MAP) data at the time of kidney biopsy. We reviewed any administered antihypertensive drugs, such as calcium channel blockers, beta blockers, angiotensin converting enzyme (ACE) inhibitors, and angiotensin II receptor blockers (ARBs), and obtained information about administered immunosuppressant agents. Patients underwent regular check-ups at intervals of 3 to 6 months, and laboratory data was collected at every visit. We determined the eGFR using the Chronic Kidney Disease Epidemiology Collaboration equation [21]. Proteinuria was assessed by 24 h urine collection. We reviewed the pathological findings from the kidney biopsies and specifically checked the pathological severity in patients with IgAN, according to the Oxford classification [22].

### 2.3. Outcome Measures

In the retrospective study to evaluate long-term kidney function, we analyzed within-group differences in eGFR over time, and the mean annual rate of eGFR decline between the MHC, MGA, and IgAN groups. The date of kidney biopsy was defined as the start of the follow-up period (baseline). 

In the prospective study to evaluate mitochondrial injury, we measured the urinary copy numbers of the mtDNA genes cytochrome c oxidase subunit III (*COX3*) and mitochondrially encoded NADH dehydrogenase 1 (*MT-ND1*) in the three groups. 

### 2.4. Urinary mtDNA Copy Number Quantification

Urinary copy numbers of the mtDNA genes *COX3* and *MT-ND1* were measured by real-time quantitative polymerase chain reaction (RT-qPCR). DNA was isolated from urine samples (1.75 mL) using DNA isolation kits from Norgen Biotek (Thorold, ON, Canada; cat. no. 18100). We analyzed DNA concentrations using a NanoDrop Spectrophotometer (Thermo Fisher Scientific, Waltham, MA, USA). RT-qPCR was performed using 20 ng of template DNA and *MT-ND1* (forward: 5′-AGTCACCCTAGCCATCATTCTACT-3′, reverse: 5′-GGAGTAATCAGAGGTGTTCTTGTGT-3′) or *COX3* (forward: 5′-AGGCATCACCCCGCTAAATC-3′, reverse: 5ʹ-GGTGAGCTCAGGTGATTGATACTC-3ʹ) primers obtained from Thermo Fisher Scientific. The PCR conditions were 95 °C for 10 min, then 40 cycles of 95 °C for 15 s and 60 °C for 60 s. The mtDNA copy numbers were normalized to the TaqMan® Copy Number Reference Assay RNase P (nDNA; cat. no. 4403326; Thermo Fisher Scientific), using human genomic DNA for the standard curve. The mtDNA copy numbers were calculated using Copy Caller software (Thermo Fisher Scientific, Carlsbad, CA, USA) and expressed as mtDNA/nDNA ratios [11].

### 2.5. Statistical Analyses

Descriptive characteristics of the study population are reported as the mean ± standard deviation for continuous variables and as the frequency count with percentage for categorical and binary variables. Comparisons of differences between groups were performed using either the Student’s *t*-test or Mann–Whitney test for continuous variables and either a χ^2^ test or Fisher’s exact test for categorical variables, as appropriate. All statistical tests were two-sided and the results are presented with 95% confidence intervals. We analyzed the statistical significance of time effects to determine whether changes in eGFR were meaningful, and evaluated within-group differences using a linear mixed model. The following variables were adjusted: age, sex, DM, HTN, eGFR, SBP, and DBP. All *p* values less than 0.05 were considered statistically significant. Statistical analyses were performed using SPSS version 25 for Windows (SPSS Inc., Chicago, IL, USA).

## 3. Results

### 3.1. Comparison of Long-Term Kidney Function Among the Three Groups in the Retrospective Study 

#### 3.1.1. Long-Term Kidney Function in MGA Group Was Worse than in MHC Group

SBP and MAP were higher in the MGA group than in the MHC group, while other demographic and clinical variables were not significantly different between the groups (Table 1). The mean follow-up duration was longer in the MHC group. The mean rate of eGFR decline was significantly higher in the MGA group compared with the MHC group (Table 2). There were no significant differences in mean eGFR between the two groups from baseline to 3 years later; however, after that the mean eGFR was significantly lower in the MGA group, for up to 10 years after baseline (Appendix A). The time × group effects of the eGFR were significantly different between the two groups (Figure 2).

#### 3.1.2. Long-Term Kidney Function in MGA Group was Comparable to that in IgAN Group

The mean DBP was lower in the MGA group compared with the IgAN group, although not significantly. More patients in the IgAN group were treated with ACE inhibitors/ARBs than in the MGA group (Table 1). The mean rate of eGFR decline was similar between MGAs and IgAN groups (Table 2). There were no significant differences in the mean eGFR between the two groups from baseline to 9 years later; however, the mean eGFR was significantly lower in the MGA group 10 years after baseline (Appendix A). The MGA and IgAN groups did not show significant differences in the time × group effects of the eGFR (Figure 2).

### 3.2. Comparison of Urinary mtDNA Copy Numbers in the Three Groups in the Prospective Study 

#### 3.2.1. Urinary mtDNA Copy Numbers Were Higher in the MGA Group Than in MHC Group

Mean proteinuria levels were higher in the MGA group compared with the MHC group, and the mean SBP tended to be higher in the MGA group as well, although not significantly. There were no significant differences between the two groups in other demographic and clinical variables (Table 3). Log_10_*MT-ND1*/nDNA and log_10_*COX3*/nDNA ratios were significantly higher in the MGA group compared with the MHC group (Figure 3). 

#### 3.2.2. Urinary mtDNA Copy Numbers Did Not Differ between the MGA and IgAN Groups

Mean proteinuria levels were lower in the MGA group compared with the IgAN group. The mean MAP was significantly lower, and the mean DBP trended lower in the MGA group compared to the IgAN group. The mean eGFR was significantly higher and mean proteinuria was significantly lower in the MGA group compared with the IgAN group. More patients in the IgAN group were treated with ACE inhibitors/ARBs than in the MGA group (Table 3). There were no significant differences between the MGA and IgAN groups in log_10_*MT-ND1*/nDNA and log_10_*COX3*/nDNA ratios (Figure 3). We recruited patients with MGAs and IgAN who were treated with angiotensin II receptor blockers (ARBs) or angiotensin converting enzyme (ACE) inhibitors in the prospective study and analyzed the differences in urinary mtDNA levels between the two groups. There were no differences in urinary mtDNA levels between the two groups (Appendix A). Although the difference was not statistically significant, endocapillary proliferation, and segmental sclerosis tended to be higher in the IgAN group of the prospective study compared with the retrospective study (Appendix A). Urinary mtDNA levels were not significantly different between the MGAs group and patients with IgAN without endocapillary proliferation and crescent formation (Appendix A). The mean baseline urinary log_10_*MT-ND1*/nDNA and log_10_*COX3*/nDNA levels in the MGAs group were 5.50 ± 0.48 and 5.45 ± 0.48 copies/μL of urine/nDNA, respectively. The MGAs group was classified into the low and high mtDNA subgroups, according to baseline urinary mtDNA levels based on a cut-off of the mean value. The higher mtDNA subgroup showed a lower MAP and a trend of higher eGFR and lower proteinuria levels compared with the lower mtDNA subgroup (Appendix A).

### 3.3. Analysis of All Patients with MGAs Included in the Retrospective and Prospective Studies 

Of the 1453 patients who underwent a kidney biopsy during the planned study, 91 were diagnosed with MGAs (6.3%). Most of the patients were men (67.2%) and the mean age at diagnosis was 30.7 ± 14.4 years. The mean eGFR was 99.5 ± 17.3 mL/min/1.73 m^2^ and 10.9% of the patients had HTN at diagnosis. Half of patients underwent kidney biopsies due to the concomitant presence of hematuria and proteinuria. Isolated hematuria and proteinuria were the reason for kidney biopsies in 17.2% and 32.8% of the patients, respectively. Blood pressure was higher than that of the matched healthy controls, but lower than that of the IgAN group. Hematuria was found in 67.2% of patients with MGAs and the mean urinary red blood cell counts using a high-power field were 27.6 ± 38.1, which was similar to the IgAN group. Although the amount of proteinuria were lower than among the IgAN group, proteinuria was observed in 82.8% of patients with MGAs. Immunosuppressive treatment was provided to 20.3% of patients enrolled in the study (Appendix A). 

## 4. Discussion

To our knowledge, there has been no study evaluating long-term kidney function and the urinary mtDNA levels in MGA. In this study, we retrospectively evaluated long-term kidney function and prospectively analyzed urinary mtDNA copy numbers in patients with MGAs compared with MHCs and patients with IgAN. The prevalence of MGAs among patients undergoing kidney biopsy in our center over the last 17 years was 6.6%, consistent with their prevalence in the COBRA cohort (5.4% (19/356)), which involved three medical centers over 2 years [1]. MGAs were associated with deterioration of kidney function and elevated urinary mtDNA copy numbers.

We analyzed differences in long-term kidney function by measuring the time × group effect of the eGFR and the mean rate of eGFR decline. The time × group effect differed between the MGA and MHC groups, but not between the MGA and IgAN groups. The mean rate of eGFR decline in patients with MGAs was higher compared with MHCs and was not different with patients with IgANs. These results indicate that long-term kidney function in patients with MGAs is worse than in healthy individuals and comparable to that in patients with IgAN. Since IgAN is the most common cause of primary GN worldwide, we used IgAN as a representative GN to assess the long-term kidney function of MGAs. Future studies are needed to compare MGAs with other types of GN, especially non-proliferative GN including hypertensive nephrosclerosis and secondary glomerulosclerosis. Overall, the findings indicate that MGAs, which are believed to be benign, may actually result in the deterioration of kidney function in the future. However, there were some limitations in these analyses. First, because these analyses were performed retrospectively, the follow-up duration differed between groups. In particular, a large number of patients with MGAs were excluded because of short follow-up duration. One-third of these patients entered the army after kidney biopsy, as they were initially referred to our center due to proteinuria and/or hematuria observed during their conscription examinations. These patients were relatively young and did not have underlying disease; therefore, there may be selection bias due to the exclusion of patients expected to maintain relatively good kidney function in the future. Another reason for the follow-up loss in the MGA group could be that many patients in this group maintained good kidney function at early phases of follow-up and did not return for subsequent appointments because they deemed it unnecessary. As a result, it is possible that long-term kidney function was worse in the MGA group as a whole because only patients with poor or deteriorating kidney function were followed-up for long periods. Second, there are limits in the methods used to evaluate long-term kidney function. In clinical trials, differences in long-term kidney function are typically measured by comparing the incidences of death and end-stage kidney disease, with serum creatinine doubling or 40% decline in eGFR used as the primary end point. Because the eGFRs of the MHCs and patients with IgAN were matched to those of patients with MGAs, their kidney functions were well preserved at baseline. This made it difficult to use the aforementioned primary end points, so the eGFR time × group effect and the mean rate of eGFR decline were used instead. As mentioned earlier, the former is limited by differences in follow-up duration and selection biases, and the latter remains a controversial method of assessing long-term kidney function [23].

Our results demonstrated that urinary *MT-ND1* and *COX3* copy numbers correlated well with each other and were higher in the the MGA group than in MHC group, but did not differ between the MGA and IgAN groups. In the retrospective study, there were no difference in baseline eGFR and proteinuria levels between the MGA and IgAN groups. However, the IgAN group had lower baseline eGFR and higher proteinuria levels than the MGAs group in the prospective study. Previous studies have suggested that the presence of endocapillary proliferation and crescent formation among the MEST-C scores may represent particularly severe pathologic findings which correlate to poor prognosis in patients with IgAN [22,24,25,26,27]. The IgAN group in the prospective study tended to have more patients with endocapillary proliferation and segmental sclerosis than those in retrospective study. Since MGAs are characterized by few pathological changes, we compared patients with MGAs and IgAN without endocapillary proliferation and crescent formation in urinary mtDNA levels. There were no differences in urinary mtDNA levels between the two groups. These results suggest that mitochondrial injury is associated with MGAs, and that although there was no clear evidence of kidney damage in pathological findings, renal cell damage may exist in patients with MGAs. Mitochondrial injuries are involved in the pathogenesis of various kidney diseases. Recently, we demonstrated that mitochondrial injury is associated with IgAN pathogenesis by measuring urinary mtDNA levels; similarly, this study suggests that MGA pathogenesis involves mitochondrial injury as well. Interestingly, when dividing the MGAs group into two subgroups according to urinary mtDNA levels, the subgroup of patients with higher mtDNA had a lower MAP and a trend of higher eGFR and lower proteinuria levels. These findings are consistent with our previous findings in patients with IgAN [17]. These results may indicate that mitochondrial damage is involved in the early stage of glomerular inflammation. Further studies are required to support this hypothesis. We could not determine whether urinary mtDNA may serve as a valuable biomarker for predicting the prognosis of patients with MGAs throughout the current study. In future studies, if changes in urinary mtDNA levels, using the follow-up urine sample from patients with MGAs, are measured and compared with changes in clinical variables including eGFR, the value of urinary mtDNA levels as potential prognostic markers could be determined. There were some limitations in this analysis. First, it was not possible to determine the impacts of mitochondrial injury on MGA pathogenesis or prognosis, as changes in urinary mtDNA levels were not measured during the follow-up period. To examine this, further analyses using urine samples obtained periodically during the follow-up period are currently underway. Second, we demonstrated mitochondrial injury in patients with MGAs, but we did not determine the mechanism of its formation. Previous studies have demonstrated that acquired factors associated with various kidney diseases, including oxidative stress [28], ischemia [29], and renin–angiotensin–aldosterone system activation [30,31], can lead to mitochondrial injury. Therefore, these factors may be involved in MGA pathogenesis, but experimental research is needed to investigate this. Third, the lack of systemic mtDNA levels is an important limitation. We measured urinary mtDNA levels, as previous studies have reported that urinary mtDNA levels, but not circulating mtDNA levels, are correlated with kidney dysfunction and the clinical outcomes of various kidney diseases, including acute kidney injury [32,33], chronic kidney disease [9,18], and obesity-related hyperfiltration [11]. Age-/sex-matched healthy volunteers were collected to minimize the effects of other factors that may increase circulating mtDNA levels. However, because mtDNA could be easily filtered into the urine, elevated urinary mtDNA may reflect systemic mitochondrial injuries due to other confounding clinical factors. Measurements of circulating mtDNA levels could compensate for this limitation.

MGAs were diagnosed in 6.3% of patients undergoing biopsies during the planned study period, which is consistent with previous findings [1]. MGAs were more common in young men. This result may be affected by the conscription screening system in the Republic of Korea which mandates conscription examinations for men in their 20s. Therefore, epidemiologic features of MGAs need to be confirmed in MGAs cohorts of more diverse populations in the future. We measured the albuminuria levels in only 11 of the patients with MGAs because the measurement of total urine protein is recommended for established glomerulopathies (proteinuria >500 mg/d) [34]. Nevertheless, most of the proteinuria was albuminuria, in the patients whose measurements were collected. This may be another sign of glomerular damage in patients with MGAs. Although kidney function was relatively well maintained and proteinuria was less than the IgAN group at the time of diagnosis, given the findings of poor long-term kidney function in patients with MGAs, clinicians should pay careful attention to these patients. 

The most important limitation of both the retrospective and prospective studies is the limited diagnostic accuracy of MGAs. To overcome this, a pathologist other than the one who performed the initial diagnoses reexamined the kidney tissues of enrolled patients with MGAs. Although we analyzed kidney tissue that had more than 12 glomeruli to diagnose MGAs, it is possible that the obtained kidney tissues may have been insufficient for diagnosis, as MGAs are a diagnosis of exclusion, or may have contained lesions too early in development to be specifically diagnosed as GN. For accurate diagnosis of MGAs, a repeat biopsy is needed after a sufficient time period has elapsed. Two patients enrolled in the retrospective study who displayed increasing proteinuria and deterioration of kidney function during the follow-up period had repeat biopsies and were again diagnosed with MGAs. However, most patients did not receive repeat biopsies, especially those with increased proteinuria or decreased renal function.

## 5. Conclusions

In conclusion, this study demonstrates that MGAs are associated with deteriorating long-term kidney function. In patients with MGAs, mitochondria injury was evident, despite few additional pathological changes. The results suggest that clinicians should conduct close long-term follow-up of patients with MGAs. 

## Figures and Tables

**Figure 1 jcm-09-00033-f001:**
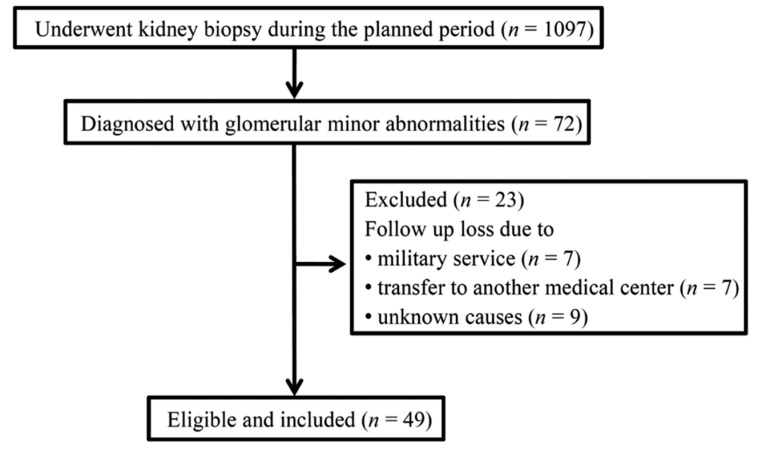
Study population for the retrospective study.

**Figure 2 jcm-09-00033-f002:**
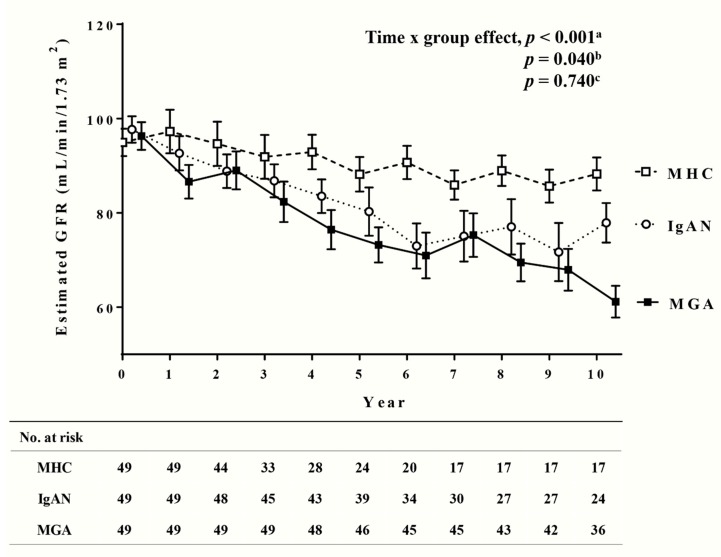
Changes in the estimated glomerular filtration rate (eGFR) in the minor glomerular abnormality (MGA), matched healthy control (MHC), and immunoglobulin A nephropathy (IgAN) groups in the 10 years after kidney biopsy. Time effects were tested using a linear mixed model. ^a^ MGA and MHC groups; ^b^ MHC and IgAN groups; ^c^ MGA and IgAN groups.

**Figure 3 jcm-09-00033-f003:**
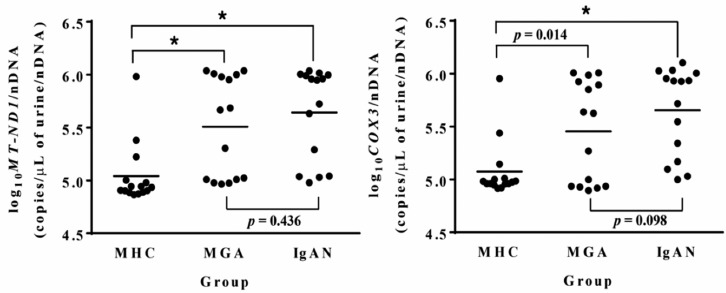
Urinary mitochondrial DNA copy numbers in the matched healthy control (MHC), minor glomerular abnormality (MGA), and immunoglobulin A nephropathy (IgAN) groups. Data were analyzed by the Mann–Whitney test. * *p* < 0.001. *MT-ND1*, mitochondrially encoded NADH dehydrogenase 1; COX3, cytochrome c oxidase subunit III.

**Table 1 jcm-09-00033-t001:** Baseline characteristics of the MGA, MHC, and IgAN groups in the retrospective study

Variable	MHC Group (*n* = 49)	MGA Group (*n* = 49)	*p* Value ^a^	IgAN Group (*n* = 49)	*p* Value ^b^
Age (years)	29.63 ± 12.95	29.18 ± 13.22	0.866	29.08 ± 11.35	0.967
Sex (male)	31 (63.3%)	31 (63.3%)	>0.999	31 (63.3%)	>0.999
Body mass index (kg/m^2^)	22.86 ± 4.43	24.24 ± 4.33	0.139	23.23 ± 3.82	0.222
Diabetes	0 (0.0%)	0 (0.0%)	-	1 (2.0%)	>0.999
Hypertension	0 (0.0%)	3 (6.1%)	0.242	8 (16.3%)	0.110
Systolic blood pressure (mmHg)	119.24 ± 9.12	125.10 ± 11.20	**0.006**	126.12 ± 12.68	0.674
Diastolic blood pressure (mmHg)	75.37 ± 6.93	77.96 ± 10.60	0.160	82.43 ± 8.42	0.076
Mean arterial pressure (mmHg)	88.08 ± 14.83	93.67 ± 9.91	**0.032**	96.33 ± 8.99	0.168
Baseline SCr levels (mg/dL)	0.98 ± 0.19	0.97 ± 0.21	0.881	0.97 ± 0.21	>0.999
Baseline eGFR (mL/min/1.73 m^2^)	97.04 ± 17.41	98.13 ± 18.30	0.761	98.48 ± 17.31	0.924
Baseline proteinuria (mg/day)	ND	858.3 ± 799.3	ND	856.8 ± 689.3	0.993
Use of ARB or ACE inhibitors	ND	23 (46.9%)	ND	39 (79.6%)	**0.001**
Use of immunosuppressant	ND	12 (24.5%)	ND	11 (22.4%)	0.812

Data are expressed as the mean ± standard deviation for continuous variables and *n* (%) for categorical variables. ^a^ MGA and MHC groups; ^b^ MGA and IgAN groups. MHC, matched healthy control; MGA, minor glomerular abnormality; IgAN, immunoglobulin A nephropathy; SCr, serum creatinine; eGFR, estimated glomerular filtration rate; ARB, angiotensin II receptor blocker; ACE, angiotensin converting enzyme; ND, not determined.

**Table 2 jcm-09-00033-t002:** Changes in eGFR in the MGA, MHC, and IgAN groups in the retrospective study

Variable	MHC Group (*n* = 49)	MGA Group (*n* = 49)	*p* Value ^a^	IgAN Group (*n* = 49)	*p* Value ^b^
Mean follow-up duration (months)	110.69 ± 20.82	64.40 ± 44.27	**<0.001**	107.80 ± 57.68	**<0.001**
Mean annual rate of eGFR decline (mL/min/1.73 m^2^/year)	0.64 ± 1.10	3.89 ± 5.95	**<0.001**	3.09 ± 6.13	0.570

Data are expressed as the mean ± standard deviation. ^a^ MGA and MHC groups; ^b^ MGA and IgAN groups. MHC, matched healthy control; MGA, minor glomerular abnormality; IgAN, immunoglobulin A nephropathy; eGFR, estimated glomerular filtration rate.

**Table 3 jcm-09-00033-t003:** Baseline characteristics of the MGA, MHC, and IgAN groups in the prospective study

Variable	MHC Group (*n* = 15)	MGA Group (*n* = 15)	*p* Value ^a^	IgAN Group (*n* = 15)	*p* Value ^b^
Age (years)	35.80 ± 11.82	35.80 ± 17.37	0.775	35.80 ± 12.55	0.683
Sex (male)	12 (80.0%)	12 (80.0%)	>0.999	12 (80.0%)	>0.999
Body mass index (kg/m^2^)	22.90 ± 1.54	25.91 ± 5.32	0.142	26.67 ± 4.80	0.892
Diabetes	0 (0.0%)	1 (6.7%)	>0.999	1 (6.7%)	>0.999
Hypertension	0 (0.0%)	4 (26.7%)	0.100	4 (26.7%)	>0.999
Systolic blood pressure (mmHg)	108.67 ± 9.15	119.73 ± 16.73	0.061	129.80 ± 15.13	0.116
Diastolic blood pressure (mmHg)	72.40 ± 8.98	69.07 ± 10.02	0.345	77.80 ± 13.02	0.067
Mean arterial pressure (mmHg)	84.49 ± 8.42	85.96 ± 10.87	0.683	95.13 ± 12.90	**0.050**
Baseline SCr levels (mg/dL)	0.92 ± 0.13	0.91 ± 0.14	0.902	1.21 ± 0.32	**0.009**
Baseline eGFR (mL/min/1.73 m^2^)	100.42 ± 11.76	103.77 ± 12.86	0.389	77.59 ± 19.02	**<0.001**
Baseline proteinuria (mg/day)	67.47 ± 22.53	563.41 ± 774.3	**<0.001**	2021.6 ± 1636.4	**<0.001**
Use of ARB or ACE inhibitors	ND	6 (40.0%)	ND	15 (100.0%)	**0.001**
Use of immunosuppressant	ND	1 (6.7%)	ND	5 (33.3%)	0.169

Data are expressed as the mean ± standard deviation for continuous variables and *n* (%) for categorical variables. ^a^ MGA and MHC groups; ^b^ MGA and IgAN groups. MHC, matched healthy control; MGA, minor glomerular abnormality; IgAN, immunoglobulin A nephropathy; SCr, serum creatinine; eGFR, estimated glomerular filtration rate; ARB, angiotensin II receptor blocker; ACE, angiotensin converting enzyme; ND, not determined.

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
