# Peer review of "Minor Glomerular Abnormalities are Associated with Deterioration of Long-Term Kidney Function and Mitochondrial Injury"

_jcm, 2019, doi:10.3390/jcm9010033_

Round 1
Reviewer 1 Report
In the article entitled "Minor Glomerular abnormalities are associated with deterioration of long term kidney function and mitochondrial injury", Yu et al investigated the renal prognosis of un ill-defined entity called minor glomerular abnormalities (MGA). Mitochondrial injury with elevated urinary mitochondrial DNA copy numbers may mirror the underlying pathophysiology which remains unclear. They identify MGA as a potential harmful pattern which impacts renal prognosis.
The authors focus on a challenging and barely documented issue. This article is well written with a precised methodology though retrospective and prospective investigations.The authors also addressed the limits of their study.
The weakness of the study is recognized by the authors at the end of the discussion and remains a major issue in this article. An accurate pathological evaluation is very important to diagnose minor glomerular abnormalities (MGA). May the author strenghten this point with the description of the pathological evaluation in the methodology section ? Were all biopsies evaluated through light microscopy, immunofluorescence and electronic microscopy ? Did the authors discard biopsies with insufficient number of glomeruli ? or did they use some threshold or criteria to exclude some biopsies? As MGA is a diagnosis of exclusion, authors should notably precise negative criteria for this diagnosis.
The authors highlight the scarcity of the studies about MGA. A total of 64 patients (49 retrospective and 15 prospective) with MGA are included in this study. I think that it would be interesting to describe the characteristics of this important and unique cohort in the result section and to present them in a supplemental table. Hematuria is not precised. Was proteinuria the only cause for kidney biopsy in MGA patients ?
During the retrospective step, could the authors precise the recruitment of matched healthy controls (MHC) ? Hematuria and baseline proteinuria are not determined in MHC in table 1. The group of MHC may present "silent" nephropathy. How MHC in the retrospective part of the work are differenciated from MGA or IgA nephropathy ?
The authors chose to compare MGA to MHC and to patients with IgA nephropathy (IgAN). The characteristics of IgAN are different in the retrospective and prospective steps of this work. In the retrospective step; baseline eGFR and proteinuria are similar between MGA and IgAN. On the contrary, in the prospective part, IgAN have higher baseline proteinuria and lower baseline eGFR. During IgAN, pathological evaluation relies notably on Oxford classification (MEST-C score) with a prognostic value. As MGA present with few pathological changes, IgAN with minor lesions (notably without crescent or endocapillary proliferation) would have been an interesting control group. Did the authors select patients with IgAN according to histological evaluation ? The authors should precise the histological characteristics of IgAN in the prospective and retrospective parts.
Page 4 line 149. "Although the difference was not statistically significant, the mean rate of eGFR decline was higher in the MGA group than in the IgAN group" should be replaced by "The mean rate of eGFR decline was similar between MGA and IgAN groups"
Line 240-241. renin angiotensin aldosterone system activation can lead to mitochondrial injury. Angiotensin converting enzyme (ACE) and angiotensin II receptor blocker (ARB) inhibitors were statistically more frequently used in IgAN patients than MGA (table 3). May the results of urinary mitochondrial DNA copy numbers be modified by the treatment ?
Reviewer 2 Report
In their retrospective analysis, Wu et al demonstrated in a collective of 49 patients that even Minor Glomerular Abnormalities (MGA) are associated with a deterioration of Long-Term Kidney Function and Mitochondrial Injury. Amazingly, the decline and Long-term kidney function in the MGA group was comparable to that in the IgAN group. This observation is of great relevance as it provides evidence that even subtle glomerular damages deserve attention and probably aquire more nephrologic support than usually thought.
Major remarks:
Since glomerular damage is the primary concern in MGA, it is essential to differentiate the proteinuria. Statements on albuminuria are currently completely missing ,allthough albumin-creatinine ratio (ACR) or urine protein-creatinine ratio (PCR) level are robust predictors of cf-mtDNA and cf-nDNA in CKD patients (see also Chang et al.: BMCNephrology (2019) 20:391). With regard to the long-term prognosis, the degrees of tubulointerstitial damage are particularly important. In addition, there is no information on the extent of vascular damage. Wu et al have already shown (ref. 16) that increased mtUrin - DNA - excretion in IgA nephropathy correlates with a worse prognosis. Figure 3 clearly shows that both groups (MGA and IgA) divide into more or less two groups: increased versus normal mtUrin-DNA excretion. In this regard, an analysis would be useful whether the MGA has also a less favorable prognosis for elevated mt urine DNA. In addition to the proliferative IgA glomerulopathy a supplementary analysis for non-proliferative glomerulopathy would be useful (for example hypertensive nephroskleorse or secondary glomerulosclerosis).Author Response
Please see the attachment.

Round 2
Reviewer 1 Report
The authors answered all my questions and modified the manuscript accordingly
Reviewer 2 Report
The authors substantially improved the manuscript.